# Influence of Work Hardening on the Surface of Backup Rolls for a 4-High Rolling Mill Fractured during Rolling Campaign

**DOI:** 10.3390/ma15103524

**Published:** 2022-05-13

**Authors:** Rumualdo Servin-Castañeda, Sixtos Antonio Arreola-Villa, Alejandro Perez-Alvarado, Ismael Calderón-Ramos, Ruben Torres-Gonzalez, Alonso Martinez-Hurtado

**Affiliations:** 1Facultad de Ingenieria Mecanica y Electrica, Universidad Autónoma de Coahuila, Unidad Norte, Monclova Coahuila 25750, Mexico; svilla@uadec.edu.mx (S.A.A.-V.); alejandro.perez@uadec.edu.mx (A.P.-A.); i.calderon@uadec.edu.mx (I.C.-R.); 2Facultad de Ingenieria Mecanica y Electrica, Universidad Autónoma de Coahuila, Unidad Torreon, Torreon Coahuila 27410, Mexico; ruben.torres@uadec.edu.mx (R.T.-G.); martinez_alonso@uadec.edu.mx (A.M.-H.)

**Keywords:** backup roll, spalling, fracture, work hardening, mechanical contact

## Abstract

Backup rolls are the main tool in a four-high rolling mill; the rolling forces applied in load cells promote the fatigue of the material due to mechanical contact between backup rolls and work rolls. This work investigated the causes of recurrent failures in backup rolls, with cracking always initiated on the surface of the roll body and finishing in the main radius between neck and roll body. Aiming to find the causes of failure, visual inspection and morphology of the fracture were performed, complemented with mechanical tests of hardness on the stress concentration area, in addition to validating the results by applying the finite element method, using ANSYS Mechanical Static Structural Software. It was concluded that the fatigue crack initiated on the surface of BUR due to work hardening continued growing up over the fatigued material, creating beach marks, and finally, a fracture occurred in the main radius of BUR due to stress concentration. The work hardening is the main cause of spalling on BURs and other mechanical components exposed to mechanical contact.

## 1. Introduction

The typical configuration of a four high rolling mill comprises four rolls: two work rolls (WRs) in contact with the strip to deform it and two backup rolls (BURs) supporting and transmitting the rolling forces [1]. BURs are normally produced of steel 5% Cr and quenched and tempered on the surface, obtaining a layer of hard material that is in contact with the WR. Strip flatness requires a strict control of pressure between BURs and WRs because it is the basis of strip shape quality; BUR has an important role because it needs to be wear-resistant on the barrel but also tough enough to support the flexion and rolling load. Different combinations of mechanical crowns to distribute the pressure along the mill campaign have been studied using mathematical models and numerical models employing the finite element method (FEM). Smart crowns in cold and hot rolling mills [2,3] have been studied to reduce wear in work rolls and compensate the pressure distribution to reduce spalling. Negative crowns in work rolls were employed by Wang et al. [4] to evenly distribute the contact stress between backup and work rolls. Wang et al. and Cao et al. [5,6] studied continuous variable crowns in the backup rolls and the work rolls in hot rolling and found an improved strip profile as well as reduced wear. Liu et al. [7] established a systematical roll contour configuration scheme for four-high CVC mills considering the axial force generated. The geometry of the backup roll was optimized to reduce the axial force magnitude. However, all researchers converge to the same idea, that the stress concertation zone is near the end of the barrel of the BUR, which is influenced by all process rolling variables combined with different types of crowns to control the strip flatness and spalling due to roll wear caused by non-uniform pressure distribution. Therefore, it is necessary to analyze the effect of mechanical contact at the stress concertation zone and determine the impact of work hardening.

The control of pressure on the strip and the contact surfaces of backup and work rolls is one of the most important problems in the rolling process. BURs are essential to overcoming this because they bend to support the rolling forces. The non-uniform pressure distribution generates localized loads that develop mechanical fatigue. The analysis of the pressure distribution on the strip width in cold [8] and hot [9] rolling shows peaks of pressure near the ends of the barrel of backup rolls. Kong et al. and Cao et.al. [10,11] determined that the pressure distribution is related to the strip width and can be modified by the chamfers at the ends of the barrel of backup rolls.

When pressure is not distributed adequately, the process generates zones with high stress concentration. Dong et al. studied a spalling in a wide and thin strip hot rolling process. Their conclusion was that roll wear had a significant impact on contact stress distribution; mainly, wear contours of WR led to produce huge contact stress peaks in two sides of the rolls [12]. The areas of high stress concentration are zones exposed to work hardening due to Rolling Contact Fatigue. Hardening occurs due to the formation of dislocations when the material is deformed in the plastic region. The continuous work of the material progressively increases the density of dislocations, interfering with their own motion. The result is increased yield strength but reduced ductility. The increase in hardness makes the material more brittle and becomes critical when the hardness is not homogeneous, causing a point of concentration of mechanical stress.

Qin et al. [13] used two rings of forged steel Cr5 to study the hardening mechanism on zones with high rolling contact fatigue. Choudhary et al. [14] studied the relationship between tensile stress–strain and work hardening behavior, describing that there are variations in work-hardening parameters with temperature and strain rates. The phenomenon can be applied to the steady-state limit of creep and is put on a physical foundation of dynamic recovery of material [15]. Work hardening is the result of rolling contact fatigue; this matter has been studied over the past several decades, and many researchers have studied the failure and microstructural alterations in bearings such as dark etching regions and white etching bands and how operating conditions affect elements such as pressure, temperature and running time; different methods have been employed to investigate the effect of microstructure alterations induced by fatigue on the hardness change [16,17,18,19,20,21]. The surface contact between BUR and WR generates mechanical fatigue because of the long periods of continuous work (28–42 days), allowing surface inspection only during mill shutdowns.

When there is no control in the distribution of pressure, the increase in hardness and fatigue of BURs, it could cause an accident. Accidents in rolling mills need to be avoided as much as possible because they promote shutdowns of the rolling mill, causing economic losses, and thus decreasing the operativity of the rolling mill. A spalling on rolls stops the mill operation for several hours, and it is more critical when it occurs on BURs because a BUR weighs more than 20 tons, depending on the design of the mill. For this reason, the failure analysis of spalling is very important. This study aims to increase the understanding of the spalling due to work hardening of BURs during the rolling process based on in situ measurements and observations. Visual inspection and morphology of the fracture were performed, complemented by mechanical tests of hardness on the stress concentration zone (confirming the location of the zone by FEM) to investigate the mechanism of the fracture. 

## 2. Methods

In this case, the BUR was produced of forged steel, surface hardened. The tempered layer has a range of original hardness from 530 HV to 580 HV. This mechanical property remains from the surface of BUR to interface of quenched and tempered material. Figure 1 shows that the spalling occurred in a four-high rolling mill, the BUR was installed in stand number four, top position, the fracture occurred close to the end of campaign, and after 28 days of continuous work, the production of strip was 178,723 tons. The actual diameter when the spalling occurred was 1219.20 mm.

### 2.1. Physical Causes

The purpose of making a preliminary examination of the failed pieces by means of fractography was to analyze the fracture features and attempt to relate the topography of the fracture surface to the causes and determine the physical failure mechanisms and how the failure happened; to understand the basic mechanisms of fracture, the conditions and characteristics of the working environment were observed, considering the status of mechanical components that were directly interacting with the failed piece [22,23].

### 2.2. Mechanical Testing of BUR

A preliminary hardness test was carried out on the surface of BUR, along the barrel in positions parallel to axis of BUR. The inspection was carried out every 101.6 mm on three lines, dividing the circumference of the barrel every 120° (see Figure 2). A total of 48 hardness tests were performed (16 tests along each line). Additionally, two metal samples were extracted, one of them called Sample 1, from the area with the lowest hardness located in the middle of the barrel, and the other one called Sample 2, from the zone with the highest hardness located at 208 mm from the operation side end of the barrel. The samples were situated in line with the keyway axis. The diagram of location of the samples and hardness inspection is shown in Figure 2. The microhardness profile was measured by a Vickers hardness test machine (Buehler Wilson VH3300, Lake Bluff, IL, USA) [24].

Two metal samples for microstructure studies were prepared and etched with 5% nital solution; the objective was to detect the microstructure changes of roll material due to mechanical contact. Additionally, for these samples, a Leica optical microscope (Leica DM750, New York, NY, USA) was used to observe the etched region of the sections [25].

### 2.3. Simulation with FEM

The distribution of pressure along the barrel of BUR is directly related to the dimensions of rolls and support points and the application of rolling loads. A simulation considering static loads including the backup roll, work roll, and strip was performed to obtain the stress distribution and relate the stresses to the fracture zone. Figure 3 shows the diagram of a four-high rolling mill.

The analyses of distribution of stress of BUR for the rolling process are based in an FEM, using the ANSYS Mechanical Static Structural software (ansys2022R1, Ansys Inc., Canonsburg, PA, USA), considering the geometrical dimensions of mechanical components shown in Table 1, the conditions of the rolling process shown in Table 2, and the mechanical properties of components involved in the process shown in Table 3. The rolling (Rf) and bending (Bf) forces are uniformly distributed and applied in the bearing areas, as shown in Figure 3. Only elastic deformation was considered for the rolls and strip by means of a model with a linear relationship of stress–strain. General joints in the lateral face of the rolls are considered, which only allow for vertical movement in the direction of force application. The lower face of the strip was fixed. The mesh was generated by ANSYS using intelligent meshing with an element size of 0.075 m. The mesh element used for all the simulations was SOLID186, an element defined by 20 nodes with three degrees of freedom per node: translations in the nodal x, y, and z directions. This element can take a hexahedral, tetrahedral, pyramid, or prism shape in function of the complexity of the geometry. The mesh had a total of 1,720,489 nodes and 765,467 elements. The contact region mesh (BUR-WR and WR-strip) is refined using 20 layers with a growth rate of 1.1; for the contact region, a bonded system (no sliding or separation between faces or edges) is implemented without penetration. The data considered for the analysis are those established for stand M-4 (Lewis United, PA, USA) of a hot rolling mill, in which the materials of the rolls are 5% Cr forged steel for BUR and nodular iron indefinite chilled double pour (ICDP) for the WR, rolling a structural steel [26].

## 3. Results and Discussion

### 3.1. The Surface Morphology

Analyzing the general panorama of Figure 1, a specific zone was detected, as shown in Figure 4. The crack origin was detected by visual inspection, and it is shown with a Chevron mark at 208 mm from the barrel end. Additionally, in this area, we can observe a series of beach marks relatively smoothed upwards; this indicates the direction of propagation of the crack. We have drawn arrows on the surface to indicate the growth of the crack due to fast fracture advancing in the direction opposite to the rotation of the BUR. This zone exhibited high hardness due to mechanical contact by stress and pressure concentration; also note the increase in surface roughness at the end of the barrel of the BUR because, in this area, the hardness is lower increasing the toughness of the material.

### 3.2. The Mechanical and Metallurgical Properties

Surface microhardness measurements were carried out along the barrel of the BUR. As mentioned in Section 2.2, the tests were performed on three lines. The average of the three measurements for each position was calculated, and the results are shown in Figure 5. We can observe the hardness ranging from 545 to 630 HV. The hardness profile indicates that there is an increase in hardness at the ends of the barrel located only 5 mm far away from the Chevron mark. Moreover, the hardness measured as a function of depth in radial direction for the two extracted samples is shown in the graphs of Figure 6. The results show an increase in hardness in both samples, but this is bigger in Sample 2. This indicates that, in both cases, there was work hardening, but it was 51 HV greater and 0.6 mm deeper in sample 2. This means that at a 1.4 mm depth, the hardness of sample 2 had values above of 580 HV, while the maximum value for sample 1 was 579 HV. The layer of work hardening is 1.8 mm for sample 2 and 1.2 mm for sample 1; both curves in Figure 6 show that the original hardness is 542–546 HV and will remain homogeneous until the interface of tempered and non-tempered material.

The material of BUR is a forged steel quenched and tempered on the surface of a barrel, which has the chemical composition and mechanical properties shown in Table 4. The composition was determined by optical emission spectrometry with SPECTROMAX LMM14 (Spectro, Kleve, Germany). The sample for chemical composition was obtained after spalling occurred, at 208 mm from the operation side end of the barrel. The mechanical properties were obtained from Servín et al. [1].

Metallurgically, the microstructure of the forged material 5Cr slightly changed, as can be seen in Figure 7a for sample 1 and Figure 7b for sample 2. In both cases, the microstructure consists of bainite laths and martensite lamella, involving some amounts of carbides. The main difference between them is the quantity and size of precipitating carbides. In Figure 7b, the size of the carbides is smaller than in Figure 7a, but they are greater in quantity due to the precipitation of carbides in the fatigue zone. 

The changes in hardness in the fatigued material are shown with microstructural alterations such as dark etching regions and white etching bands, which are due to rolling contact fatigue. In Figure 8a,b we can see the width of the etching bands for samples 1 and 2, respectively. It is wider for sample 2 because the radial depth of work hardening and hardness values are greater than those for sample 1. 

### 3.3. Stress Distribution Analyzed by FEM

The profile of stress distribution along the barrel of the BUR, applying the FEM method, is shown in Figure 9. We can see stress concentration in both sides at the ends of the barrel of the BUR, which is 323.21 MPa on the operation side and 268.17 MPa on the drive side.

In the middle of the barrel, the stress is at minimum value (130.27 MPa). An illustration of minimum and maximum stress distribution is given in Figure 10, obtained by the simulation of the rolling process by the ANSYS Mechanical Static Structural software. The results of the simulation confirm that the operation side presents more stress concentration than the drive side. Additionally, in this image, we can see the stress concentration at the main radius of both sides, located at the union of the barrel with the necks. 

## 4. Discussion

Fracture is a basic failure of mechanical fatigue due to work hardening in combination with stress concentration, or stress risers, as they are frequently known. The stress distribution profile shown in Figure 9 and the surface hardness profile shown in Figure 5 are very similar, with peaks of local stress on the ends of the barrel. The maximum hardness value of 630 HV and the maximum stress value of 323.21 MPa are located 203 mm from the end of the barrel in the operation side of BUR, consistent with previous studies that determined that contact pressure concentration exists between work and backup rolls; normally, it is located close to the end of the barrel length [1,2,3,4,5,6,7], creating a peak of maximum pressure concentration, such as is described by Liu et al., Zhang et al., Kong et al., and Cao et.al. [8,9,10,11,12]. With the application of mechanical tests, we could determine that hardness was affected in the radial direction, reaching values of 580 HV at 1.4 mm depth, which was confirmed with images of the dark etching region and white etching bands, such as is described by some researchers that have studied the alterations of areas exposed to mechanical contact in bearings and other mechanical components [16,17,18,19,20,21], as well as the metallurgical change in microstructures, increasing the size and quantity of carbides. These parameters exhibit the presence of work hardening due to contact fatigue rolling, according to a hardening mechanism induced by rolling contact described by Qin et al. [13], creating the beginning of a surface crack on the fatigue zone of a BUR, growing along the fatigued material by fragile fracture, and finishing with a spalling in the main radius located in the union of the barrel with the neck of the operation side, according to a morphology analysis, supported and confirmed by the FEM simulation using the ANSYS Mechanical Static Structural Software, where we can observe two points of stress concentration located on the end of barrel length and the main radius, which are the beginning and end of the crack, respectively. Additionally, the BURs work at 100–175 RPM for 28 days (22 h per day), resulting in approximately 3.7 × 10^6^ to 6.5 × 10^6^ cycles. This range is within the finite life range reported for a similar material by Serbino and Tschiptschin [27].

It is impossible to avoid work hardening on BUR due to rolling contact. This could be critical in combination with stress concentration and metallurgical change, because the work hardening concentrator is greatly increased. There is always a latent risk of an operational accident, yet such risk can be reduced with the control of surface hardness and maintenance of BUR after each rolling campaign. When there is no hardness control, the work hardening of BUR can generate micro-cracks, which could cause a spalling such as that analyzed in this research. To solve this failure and prevent the fracture, it is necessary to remove the fatigued material until hardness is homogeneous; it must be less than the original hardness of the surface. It is variable for each particular case of four-high rolling mill. The original hardness is configured for the tempered layer of the material that constitutes the BUR, from surface of BUR to the interface of quenched and tempered material; in this case, it is from 530 to 580 HV.

## 5. Conclusions

Mechanical fatigue is progressive and localized structural damage that occurs when a material is subjected to changes in hardness due to mechanical contact. If the local stress is high enough, this leads to the initiation of a crack, the growth of the crack, and finally, fracture.

The hardness obtained after spalling was bigger than the original hardness, and the BUR had two zones of fatigued material due to mechanical contact; one of them was the zone where the spalling started.

This research promotes the monitoring of work hardening to prevent fractures of BURs in four-high rolling mills or reduce the incidence of failures, extending the operativity of rolling mills.

## Figures and Tables

**Figure 1 materials-15-03524-f001:**
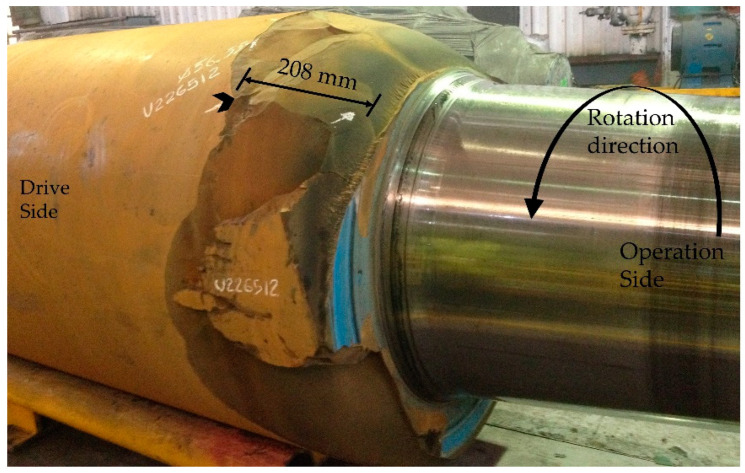
Spalling of bottom BUR 5Cr, installed in 4-high rolling mill.

**Figure 2 materials-15-03524-f002:**
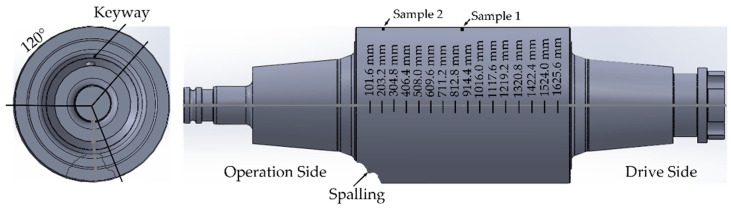
Schematic of identification and location of samples.

**Figure 3 materials-15-03524-f003:**
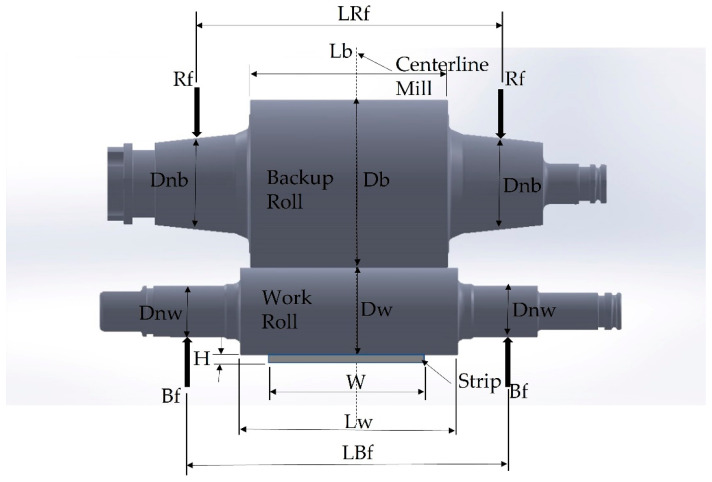
Schematic of main dimensional variables for four-high rolling mill.

**Figure 4 materials-15-03524-f004:**
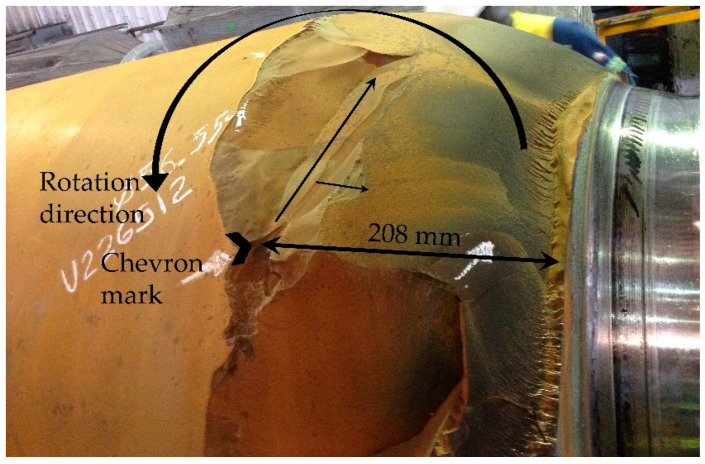
Evolution of morphology and propagation of crack.

**Figure 5 materials-15-03524-f005:**
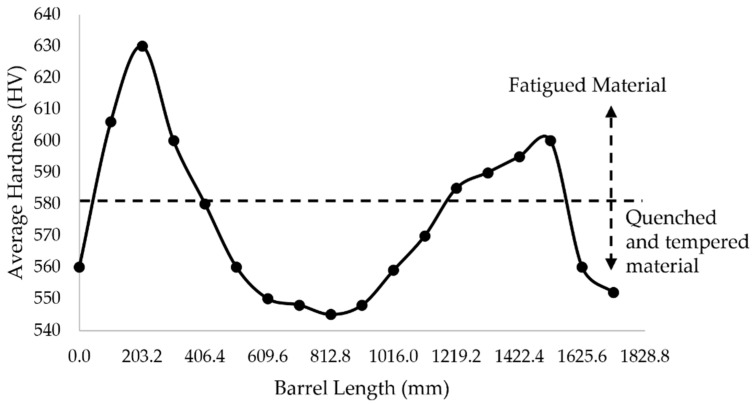
Surface hardness profile for the barrel of BUR.

**Figure 6 materials-15-03524-f006:**
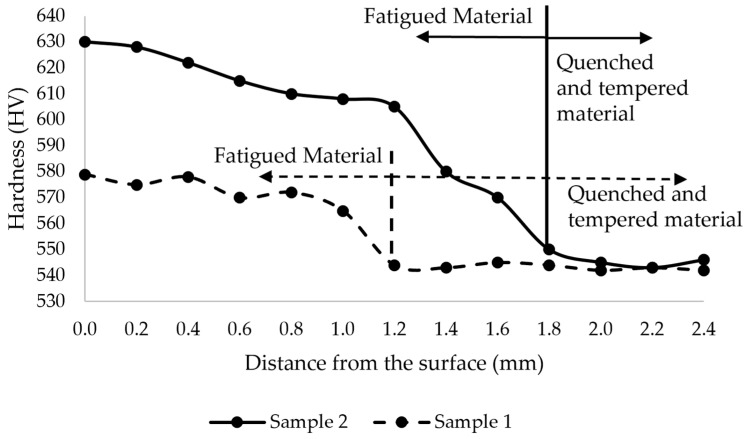
Hardness profile as function of depth for the extracted samples.

**Figure 7 materials-15-03524-f007:**
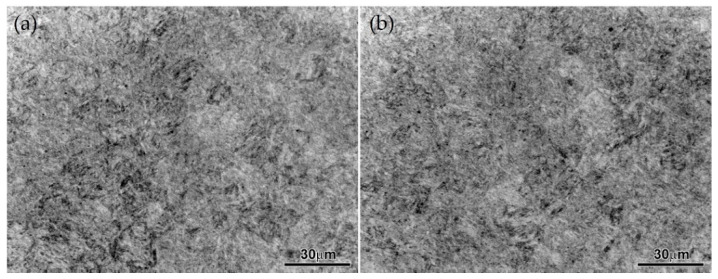
Microstructure at 500× etched with nital 5%, (**a**) sample 1, (**b**) sample 2.

**Figure 8 materials-15-03524-f008:**
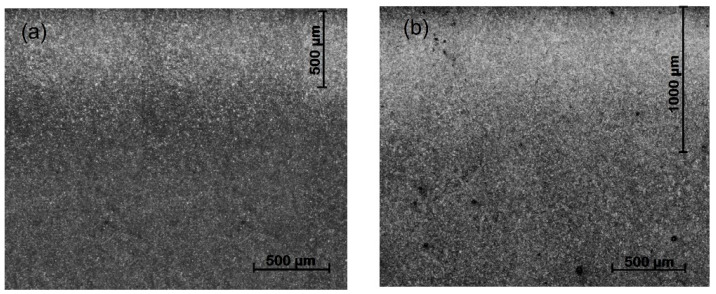
Dark etching region and white etching bands, (**a**) sample 1, (**b**) sample 2.

**Figure 9 materials-15-03524-f009:**
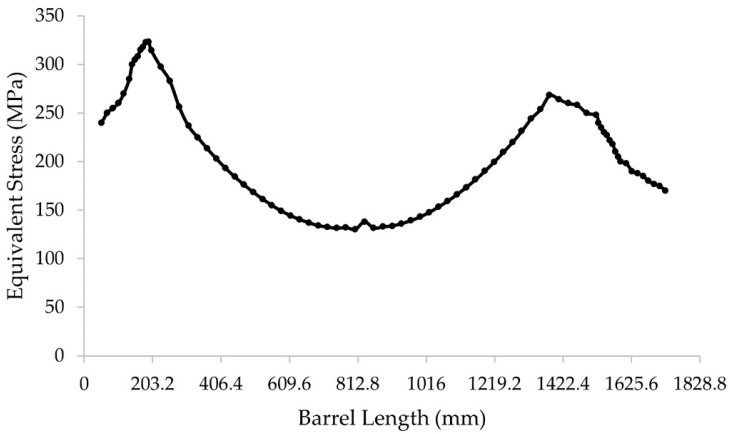
Stress profile along the barrel of the BUR.

**Figure 10 materials-15-03524-f010:**
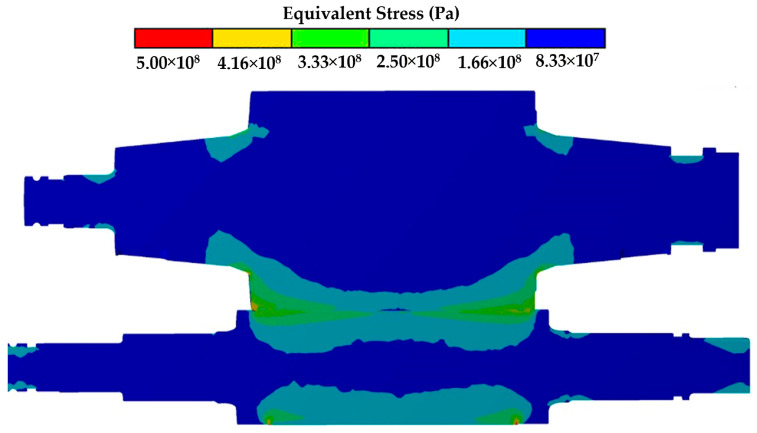
Graphical illustration of stress concentration.

**Table 1 materials-15-03524-t001:** Geometrical dimensions of components used for a 4-high rolling mill.

Mechanical Component	Value
Work roll diameter Dw	625.475 mm
Work roll barrel length Lw	1897.20 mm
Work roll neck diameter Dnw	384.175 mm
Work roll Bending force length LBf	2747.20 mm
Backup roll diameter Db	1219.20 mm
Backup roll barrel length Lb	1727.20 mm
Backup roll neck diameter Dnb	678.49 mm
Backup roll Rolling force length LRf	2717.80 mm
Strip width W	1524.0 mm
Strip thickness H	2.0 mm

**Table 2 materials-15-03524-t002:** Conditions of rolling process for a 4-high rolling mill.

Model Parameter	Value
Rolling force Rf	36,000 kN
Bending force Bf	350 kN
Work Roll Crown	0.1016 mm
Backup Roll Crown	0.0762 mm
Maximum Thermal crown WR	0.150 mm
Maximum Thermal crown BUR	0.100 mm
Maximum wear WR	0.150 mm
Maximum wear BUR	0.600 mm
Chamfer of BUR	100 mm × 1.0 mm
Chamfer of WR	100 mm × 2.8 mm

**Table 3 materials-15-03524-t003:** Mechanical properties of components for a 4-high rolling mill.

Mechanical Property	Value
Piosson’s ratio work roll	0.3
Poisson’s ratio backup roll	0.3
Young’s modulus work roll	210 GPa
Young’s modulus backup roll	220 GPa
Young’s modulus strip	210 MPa

**Table 4 materials-15-03524-t004:** Chemical composition and mechanical properties of 5% Cr Steel.

Chemical Composition (%Wt)						Mechanical Properties [1]			
C	Si	Mn	Cr	Mo	V	Yield Limit (MPa)	Hardness Range (HV)	Poisson ratio	Modulus Elasticity (GPa)
0.55	0.15	0.60	5.37	0.80	0.15	1200	530–580	0.3	220

## Data Availability

Not applicable.

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
