# Peer review of "Influence of Work Hardening on the Surface of Backup Rolls for a 4-High Rolling Mill Fractured during Rolling Campaign"

_materials, 2022, doi:10.3390/ma15103524_

Round 1

Reviewer 1 Report

This paper analyzes the causes of recurrent failure in Back Up Rolls (BURs) of the rolling mill. Through the analysis of hardness, fracture, microstructure evolution, and stress distribution at different positions, it is concluded that the cycle leads to strain hardening, microstructure evolution leads to damage, fatigue crack, and finally fracture. This is a research paper on material fatigue in practical engineering components under service conditions, and the work is meaningful. From the known knowledge, the conclusion obtained in this paper is reasonable.

However, the introduction of some analytical work in the paper needs to be explained further:

(1) It should be explained whether the mechanical properties data in Table 2 is material data or sampled from original or damaged BURs.

(2) Table 2 only gives the elastic mechanical properties data of the material, while the later analysis relates to strain hardening. The authors seem to need to explain the plastic and cyclic plastic properties of the material.

(3) It should be explained where were extracted the test samples in the BURs shown in Figures 7 and 8.

(4) In this paper, it is determined that the BUR is damaged by fatigue failure, if so the fatigue life data should be provided.

(5) For the finite element calculation of the BUR, the mesh division and finite element choice of the calculation model should be mentioned, and further mentioned whether the calculation model includes the rolled workpiece and how to consider the contact conditions in the calculation.

(6) From the context text, Figures 5 and 6 show the material hardness measured from BUR after failure, but the text also says that the original hardness of the material can be obtained in these two figures, which should be explained.

(7) According to the text, the hardness of a specific region of the BUR has changed greatly, and it is considered to be related to strain hardening. The author meanwhile points out that the microstructure of the material has changed, in Figures 7 and 8. In this way, some regions of the BUR should have plastic deformation. To this point, the authors should explain whether plastic deformation and strain hardening are considered in the finite element calculation or not.

Author Response

Thank you for your support. In the Attached file are the responses of your comments.

Best Regards 

Reviewer 2 Report

The paper does not present a significant novelty and originality. The scientific aspects are also quite poor. The manuscript consist of only 11 pages, whereas the last two pages are mainly author contributions, acknowledgments and references. The paper is more like an engineering thesis than serious a scientific paper. In my pinion the present paper is not ready for publication at the current state. VERY major corrections and adjustments are needed.

Some detailed comments are given here:

  1. There are some editorial errors, like different various spacing starting from the Line no. 77 to 87 or Line 93-98. Please revise carefully the whole paper.
  2. Caption of Table 1 is on page 4, whereas the table is on page 5.
  3. Figures presenting the characteristics have different widths – they should be unified.
  4. Including a link to researchgate is not correct for a serious scientific paper. Moreover, why there are some references underlined, i.e. Refs. 22 – 25?
  5. The Conclusions section is very poor and it consists only of two sentences. Really?
  6. The Introduction section needs revisions and corrections. A novelty and originality of your study should be highlighted which should be based on the literature review. Furthermore, figures should be omitted in this part of the paper. You should move Figure 1 to the another Section.
  7. Mixing the parameters used in FE model / simulation is not a good idea. Split the table into two or three separate tables regarding geometrical dimensions, constitutive modelling, etc You should discuss what type of material model was used and how did you asses the properties?
  8. The Section regarding FEM should be extended and corrected:
  • It is impossible to analyze details of your model including mesh size and connections of separate components of the model. It must be corrected.
  • Initial – boundary conditions must be presented in a more transparent way.
  • How did you select the mesh dimensions? Did you carry out a sensitivity study? Please comment, perform such analysis, and add the information in the paper, since the mesh size can drastically influence the results.
  • You wrote: “The mesh 130 for all the simulations had 461800 nodes and 240076 elements, with two zones of hexahedral elements with 20 nodes and tetrahedral elements with 10 nodes”. What do you meant by (..) elements with 20 nodes? The same stands for the tetrahedral elements? Please comment.
  • How did you connect hex. with tetra elements? Why did you used those two type of 3D elements? I do not see any specific reason of that. Please comment.
  • “The contact region 132 mesh is refined using 20 layers with a growth rate of 1.1”. What do you meant by that? Growth rate of 1.1?
  • What type of contact is used in SIMUFACT software?
  • What type of analysis of carried out? Dynamic / static / linear / nonlinear?

Author Response

Thank your for your support. In the attached file are the response for your comments.

Best Regards

Round 2

Reviewer 1 Report

The author carefully replied to the reviewers’ comments, and the paper’s readability was improved after being revised.

Reviewer 2 Report

The paper has been improved and can be published.